# Genetic determination and *JARID2* over-expression in a thermal incubation experiment in Casque-Headed Lizard

**Gabriel Suárez-Varón**[1], **Eva Mendoza-Cruz**[2], **Armando Acosta**[3], **Maricela Villagrán-Santa Cruz**[2], **Diego Cortez**[3]*, **Oswaldo Hernández-Gallegos**[1]

**1** Laboratorio de Herpetología, Facultad de Ciencias, Universidad Autónoma del Estado de México, Instituto Literario # 100 Centro, Toluca, Estado de México, México, **2** Laboratorio de Biología Tisular y Reproductora, Departamento de Biología Comparada, Facultad de Ciencias, Universidad Nacional Autónoma de México, Ciudad de México, México, **3** Centro de Ciencias Genómicas, UNAM, Cuernavaca, México

* dcortez@ccg.unam.mx

**Data Availability Statement:** RNA-seq data have been deposited to the NCBI-SRA database under BioProject PRJNA766022.

## Abstract

Non-avian reptiles, unlike mammals and birds, have undergone numerous sex determination changes. Casque-Headed Lizards have replaced the ancestral XY system shared across pleurodonts with a new pair of XY chromosomes. However, the evolutionary forces that triggered this transition have remained unclear. An interesting hypothesis suggests that species with intermediate states, with sex chromosomes but also thermal-induced sex reversal at specific incubation temperatures, could be more susceptible to sex determination turnovers. We contrasted genotypic data (presence/absence of the Y chromosome) against the histology of gonads of embryos from stages 35–37 incubated at various temperatures, including typical male-producing (26˚C) and female-producing (32˚C) temperatures. Our work apparently reports for the first time the histology of gonads, including morphological changes, from stages 35–37 of development in the family *Corytophanidae*. We also observed that all embryos developed hemipenes, suggesting sex-linked developmental heterochrony. We observed perfect concordance between genotype and phenotype at all temperatures. However, analysis of transcriptomic data from embryos incubated at 26˚C and 32˚C identified transcript variants of the chromatin modifiers *JARID2* and *KDM6B* that have been linked to temperature-dependent sex determination in other reptiles. Our work tested the validity of a mixed sex determination system in the family *Corytophanidae*. We found that XY chromosomes are dominant; however, our work supports the hypothesis of a conserved transcriptional response to incubation temperatures across non-avian reptiles that could be a reminiscence of an ancestral sex determination system.

## Introduction

Although squamate reptiles are the most species-rich group among non-avian reptiles (96.3% of the diversity [1]) the factors that drive variation in sex-determination mechanisms among species remain poorly understood. In squamate reptiles, sex determination occurs by two

**Funding:** This study was supported by grants from PAPIIT-UNAM (No. RA-200516 and No. RA-200518; https://dgapa.unam.mx/index.php/impulso-a-la-investigacion/papiit) and CONACyT Basic Science grant (No. 254240; https://conacyt.mx/) awarded to D.C., and UAEMéx 4668/2019SF (https://www.uaemex.mx/) to OH-G.

**Competing interests:** The authors have declared that no competing interests exist.

general methods: genotypic sex determination (GSD, the norm for Squamata) and temperature-dependent sex determination (TSD). In the first method, specific sex chromosomes control the development of the gonads, whereas in the second method, external cues, typically ambient temperature, regulate the sexual differentiation of ovaries and testis [2]. In many TSD species, ~26°C produces 100% males (i.e., male-producing temperature), whereas ~32°C produces 100% females (i.e., female-producing temperature).

Historically, researchers assumed that GSD and TSD were mutually exclusive; however, as sex determination in more non-avian reptiles has been studied, we now understand that GSD and TSD represent the endpoints of a continuum [3, 4] with some lizards showing temperature-induced sex reversal where ambient temperature can alter the genetic pathways of gonad differentiation [5]. Several squamate reptiles such as the three skinks *Niveoscincus ocellatus* (Spotted Skink) [6], *Eulamprus heatwolei* (Water-Skink) [7, 8], and *Bassiana duperreyi* (Eastern Three-Lined Skink) [4, 9], as well as the agamid *Pogona vitticeps* (Central Bearded Dragon) [10, 11] show intermediate states where the sex of the offspring is controlled by sex chromosomes under medium incubation temperatures but sex-linked genes can be overridden by alternative signaling cascades at elevated (Central Bearded Dragon [11]) or lowered incubation temperatures (Eastern Three-Lined Skink [12]).

Although mammals and birds have stable sex chromosomes, non-avian reptiles have undergone numerous sex determination changes with lineages shifting from TSD to GSD and vice versa [13]. A general explanation of why sex determination changes are more common in reptiles is lacking. Analyses of ancestral states indicated that the last common ancestor of all reptiles had TSD [14, 15] and, subsequently, GSD systems evolved in some lineages [14, 15]. It is possible, therefore, that the TSD systems we presently observe in turtles, crocodiles, and some lizards are derived from the ancestral TSD system [8, 16].

Non-avian reptiles are thought to have evolved temperature-dependent sex determination because their embryos develop in a close relationship with the environment [17]. An interesting evolutionary hypothesis suggests that species with intermediate states showing sex chromosomes and thermal-induced sex reversal may be more susceptible to sex determination turnovers [18] because the ancestral TSD system is still present, but it has been overridden by sex chromosomes [8]. This hypothesis is supported by observations in the Central Bearded Dragon (*P. vitticeps*) where TSD becomes the dominant sex determination system when the ZW chromosomes are lost [10]. Moreover, three phylogenetically distant reptiles, a turtle (*Trachemys scripta*), a crocodile (*Alligator mississippiensis*), and the Central Bearded Dragon, showed the same temperature-dependent spliced isoforms of two genes members of the family *Jumonji* of chromatin modifiers, *KDM6B* and *JARID2* [16]; *KDM6B* plays an important role in temperature-dependent sex determination signaling cascade because its knockdown results in a male-to-female sex reversal at male-producing temperatures in the turtle *T. scripta* [19]. *JARID2* has been proposed to play an important role in the temperature-dependent sex determination pathway in non-avian reptiles [20].

Pleurodonts, including iguanas, spiny lizards, and anoles have a common XY system [21, 22]. In *Anolis carolinensis* (Green Anole), researchers characterized a highly degenerated Y chromosome and a perfect dosage compensation mechanism that over-expresses the X chromosome in males [23]. In the pleurodont clade, however, Casque-Headed Lizards (*Corytophanidae*) replaced the ancestral XY system with a new pair of XY chromosomes [24, 25]. Moreover, initial observations performed in our laboratory with clutches of the Brown Basilisk (*Basiliscus vittatus*), a Casque-Headed Lizard, indicated that female-biased offspring may occur at medium-high incubation temperatures (~29°C). In this study, we explored the proximate mechanisms of sex determination in a squamate reptile, tested whether the Brown Basilisk showed temperature-dependent sex reversal. Moreover, we also explored the possibility

that *B. vittatus* exhibited genetic signatures of an ancestral temperature-dependent sex determination system.

# Materials and methods

## Animal collection

22 gravid females of *B. vittatus* were collected from a population that inhabits the tropical rainforest habitat at the community of "La Selva del Marinero" in the state of Veracruz, México (18˚26'36.3"N, 94˚37'81.9"W, ca. 170 m a.s.l). Permission for fieldwork and sampling was granted by the Secretaría del Medioambiente y Recursos Naturales (SEMARNAT Scientific Collector Permit 08–043). We sampled from April to July of 2019 and females were captured manually and with a noose. To evaluate the reproductive condition, both visual assessment and abdominal palpation were performed (females with eggs showed multiple contours in the abdomen area). Twenty-two females were confirmed to be gravid. They showed the following morphometric data (mean ± standard error): LHC = 128.1 ± 1.4 mm, and weight 59.6 ± 2.3 g.

## Laboratory conditions

Each gravid female was placed inside a separate terrarium (100 cm width x 50 cm depth x 50 cm height) until the termination of oviposition. Conditions for one female per terrarium were: thermal gradient between 20–40˚C, photoperiod of 12/12 h (photofase/scotophase), constant humidity, live insects as food, and water *ad libitum*. The terrariums were monitored daily. We obtained 130 eggs which were randomly assigned to three incubation temperatures, 26˚C, 29˚C, and 32˚C. Eggs were incubated at the three temperatures in three Percival L-30 incubators until they reached relatively late developmental stages (35–37 according to the development table established by Dufaure and Hubert [26]) at which point gonads should be differentiated. Forty-two eggs (32%) became contaminated with fungal infection and the embryos died with no correlation with the incubation temperature ($X^2$ test, $P > 0.05$). Of the 88 eggs that successfully reached the target developmental stages, 40 were used in a parallel study that aimed to establish the effect of incubation temperatures on the development of the embryos [27]. Finally, forty-eight eggs (12, 18, 18 for 26˚C, 29˚C, and 32˚C, respectively) reached the target stages 35–37. Each embryo was assigned a number so we could match genetic data versus histological data. The posterior part of the embryos, where the gonads are located, was dissected and fixed using Bouin solution for 30 minutes, washed with water for 30 minutes, and stored in 10% formaldehyde for further histological analysis. The upper part of the embryos (head and eyes) was stored in DNA/RNA shield buffer by Zymo Research (Cat. No. R1200–125) for further genetic analysis.

## Microscopy analysis

Conventional histological techniques were performed on each sample: dehydration via graded ethanol, clearing tissues in xylene, and embedding tissues in Paraplast (Sigma-Aldrich, Cat. No. 145686-99-3). Histological sections were made at five microns and stained with Ehrlich-Eosin Hematoxylin (Sigma-Aldrich, Cat. No. 17372-87-1) to facilitate the description of the structures. Samples were then viewed and imaged via a compound microscope including a digital camera.

## DNA extraction and Y-specific PCR analysis

The upper part of the embryos (head and eyes) was divided longitudinally into two fragments of equal size. We homogenized one of those fragments and collected 25 mg of tissue and then

purified DNA using the QIAamp Fast DNA Tissue Kit from QIAGEN (Cat. No. 51404). We verified the integrity of the DNA using 1% agarose gels. All DNA samples were tested for integrity (260/280 and 260/230 ratios >1.8), using a NanoDrop 2000 spectrophotometer (Thermo Scientific), and quantified in a Qubit 4 fluorometer (Thermo Scientific, MA, USA) with the Qubit dsDNA BR Assay kit from the same supplier. We confirmed the presence or absence of a Y chromosome in the samples using Y-specific primers designed previously [24]: *CAMS-AP1Y*, forward: `AGT CTC AGT CTG CAC CAG TGA AAG`, reverse: `TGA TTT CTG AGC CCA GGC AGT T`. *GOLGA2Y*, forward: `AGG CTG TCA GTC TCA CTC AGT AAG`, reverse: `CCC CAT ATT CCC AGG TTC TGT CA`. We verified that PCR reactions worked using primers against *COL1A1* (autosomal/control) forward: `TTT CGT GCA GGG TGG GTT CTT T`, reverse: `TCT GAA CTG GTG CAG CTT CAC A`. We used the Phusion Flash High Fidelity from Thermo Fisher Scientific (Cat. No. F548L) with the following program: first 98˚C - 10s, then 30 cycles of 98˚C - 2s, 66˚C - 5s and 72˚C - 10s, with a final elongation step at 72˚C - 30s. We confirmed the size of the PCR products and the presence of single amplicons in a 1% agarose gel.

### RNA extraction and RNA-seq analysis

The second fragment of the upper part of the embryos (head and eyes) was homogenized for RNA purification. In many samples, however, the RNA was degraded. For a total of six samples, we obtained sufficient good-quality RNA and generated strand-specific RNA-seq libraries (using the Illumina TruSeq Stranded mRNA Library protocol) for four samples incubated at 26˚C (three females and one male) and two samples incubated at 32˚C (two females). Each library was sequenced on Illumina HiSeq 2500 platforms at the Macrogene facility in Korea (100 nucleotides, paired-end). We generated 31,910,378 million reads on average (± 3,753,014 million reads). We reconstructed a full transcriptome using Trinity (v2.0.2, default k-mer of 25 bp) [28]. Then, RNA-seq reads from 26˚C and 32˚C samples were mapped to the reconstructed transcriptome using Kallisto (100 bootstraps) [29]. We obtained the estimated counts per transcript and we used the EdgeR package [30] to perform differential expression analyses of transcripts between incubation temperatures (26˚C versus 32˚C, and vice versa). We used the *edgeR* and *splines* R libraries, TMM (Trimmed Mean of M-values) normalization, FDR (False Discovery Rate) set at 0.0001 given the limited number of replicates at 32˚C, from which non-degraded RNA was difficult to obtain. We downloaded *A. carolinensis* (Green Anole) cDNAs and lncRNAs sequences from the Ensembl database (https://www.ensembl.org/; v.92) and we used BLASTn [31] to assign gene identities to the differentially expressed transcripts. When various transcripts for the same gene were differentially expressed, count estimates were added to obtain single values per gene. Enrichment analyses were carried out using Webgestalt (http://www.webgestalt.org/), specifying over-representation analysis, the genontology database, and the Biological Process category. RNA-seq data have been deposited to the NCBI-SRA database under BioProject PRJNA766022.

## Results

### Histological analysis showed that ovaries and testes were well-differentiated in stages 35–37 of development

To evaluate the effect of temperature on the sex of the embryo, we initially established the developmental stages where gonads were well-differentiated. We observed clear testis-specific and ovary-specific structures in stages 35–37 of embryonic development. During stage 35, the ovary showed an oval morphology (Fig 1A). The cortex and the medullary zone were well-

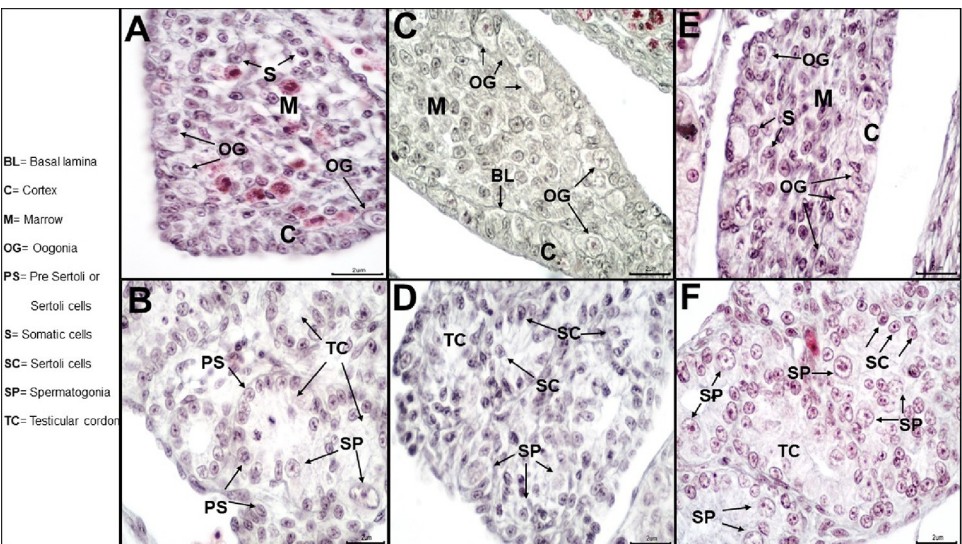

**Fig 1. Ovaries and testes at stages 35–37 of development in *B. vittatus* embryos.** a) histology of the ovary in stage 35. b) histology of the testis in stage 35. c) histology of the ovary in stage 36. d) histology of the testis in stage 36. e) histology of the ovary in stage 37. f) histology of the testis in stage 37.

differentiated. The cortex was structured by epithelial and germ cells, and the presence of some oogonia were visible. The testis in stage 35 showed cords in the medullary region together with epithelial cells, the future Sertoli cells (Fig 1B). Some of these cords presented spermatogonia in the lumen, around the medullary area. In stage 36, the ovary size increased, the cortical zone became thicker showing 2–3 layers of germ cells, closely related to somatic cells or future follicular cells in their thickest part, delimited by a basal lamina of connective tissue, in addition to a greater number of oogonia than in the previous stage (Fig 1C). Testes in stage 36 were more elongated, testicular cords increased in density in the medullary region (Fig 1D), and spermatogonia were common inside the testicular cords. Finally, in stage 37, connective tissue fibers in the ovaries delimited the cortical zone and the medullary region (Fig 1E). In the cortical region, the proliferation of oogonia covered the entire region. In stage 37 of the shape of the testicles they were elongated and curved and an entire testicle was made up of testicular cords, which were located more closely to the central area of the spinal cord. Large spermatogonia became abundant inside the testicular cords, while Sertoli cells were peripheral to the testicular cords (Fig 1F).

## No effect of incubation temperatures on the sex of *B. vittatus* embryos

We performed experiments to evaluate whether the sex of the embryos was affected by different incubation temperatures, including typical male-producing (26°C) and female-producing (32°C) temperatures. To do so, we contrasted genotypic data (i.e., presence/absence of the Y chromosome) against the histology of the gonads from stages 35–37 (i.e., presence of testicular or ovarian structures) at three incubation temperatures (26°C, 29°C, and 32°C). If *B. vittatus* presented temperature-dependent sex reversal, specific genotypes would not necessarily develop the expected gonads. For example, in a male-to-female sex reversal, individuals with a Y chromosome would develop ovaries.

We analyzed 48 embryos that reached the target developmental stages (see Methods). At 26°C we observed eight embryos with a Y chromosome and four embryos without a Y chromosome (Fig 2). At 29°C we observed 11 embryos with a Y chromosome and seven embryos

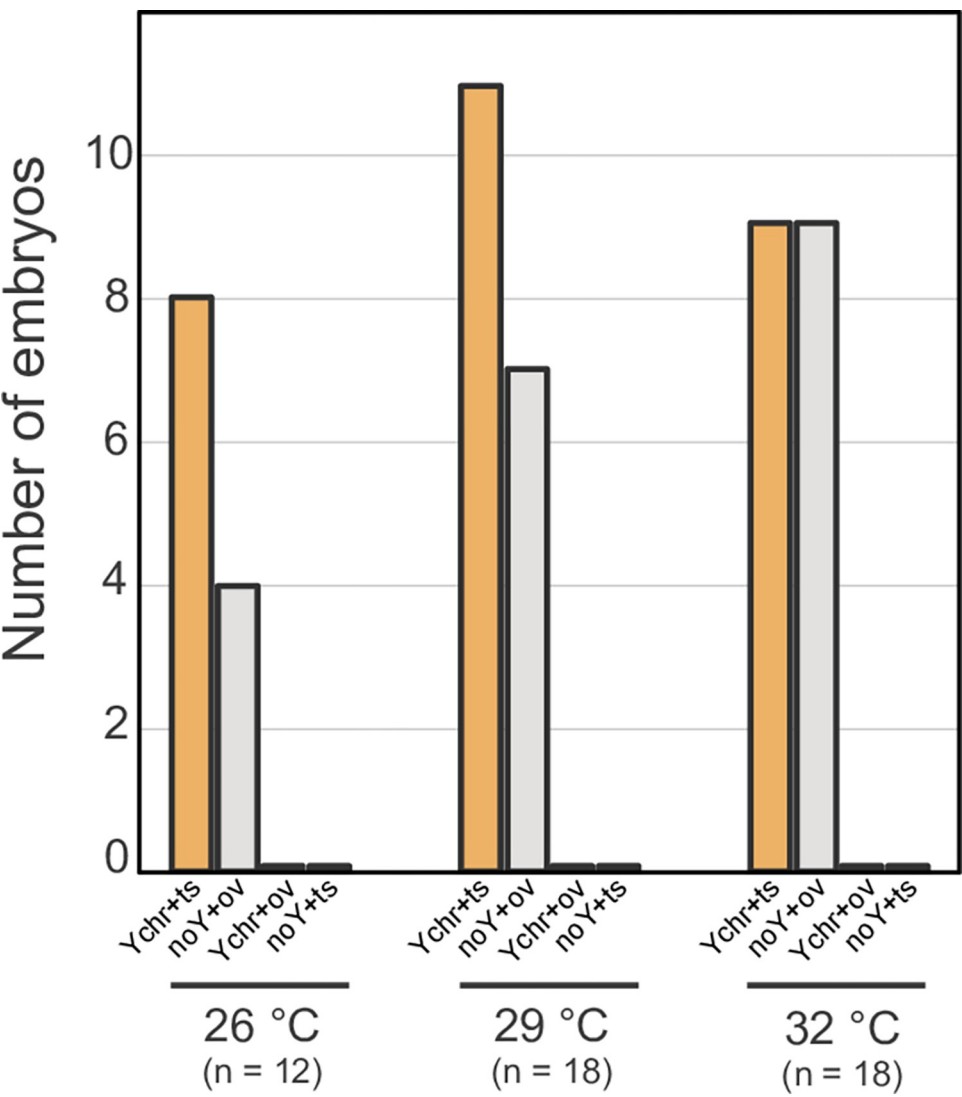

**Fig 2. Number of embryos according to the incubation temperature, genotype, and histology.** Histogram showing the number of *B. vittatus* embryos with a Y chromosome that developed testes (*Ychr+ts*; no sex reversal), the number of embryos lacking a Y chromosome that developed ovaries (*noY+ov*; no sex reversal), the number of embryos with a Y chromosome that developed ovaries (*Ychr+ov*; male-to-female sex reversal), and the number of embryos without a Y chromosome that developed testes (*noY+ts*; female-to-male sex reversal).

without a Y chromosome (Fig 2). At 32˚C we observed nine embryos with a Y chromosome and nine embryos without a Y chromosome (Fig 2). We detected the same frequency of males and females at the three temperatures ($X^2$ test, $P > 0.05$). Moreover, after contrasting the genotype of the embryos against their gonads, we found that embryos with a Y chromosome developed testes in all instances. Similarly, embryos without a Y chromosome always developed ovaries (Fig 2).

## Differential expression analysis of transcriptomic data

We performed a differential expression analysis of RNA-seq data obtained from the upper part (*i.e.*, head and eyes) of embryos incubated at 26˚C versus embryos incubated at 32˚C. We found 272 genes that were over-expressed at 26˚C and 136 genes that were over-expressed at

**Table 1. GO terms enrichment of genes over-expressed at 26˚C.**

| Gene Set | Description | Size | Expect | Ratio | P Value | FDR |
|---|---|---|---|---|---|---|
| GO:0048666 | neuron development | 1068 | 15.382 | 3.251 | 7.57E-14 | 6.88E-10 |
| GO:0010975 | regulation of neuron projection development | 475 | 6.8411 | 4.678 | 3.48E-13 | 1.58E-09 |
| GO:0030030 | cell projection organization | 1522 | 21.92 | 2.692 | 9.84E-13 | 2.98E-09 |
| GO:0120036 | plasma membrane bounded cell projection organization | 1488 | 21.431 | 2.66 | 4.33E-12 | 9.83E-09 |
| GO:0120035 | regulation of plasma membrane bounded cell projection organization | 665 | 9.5775 | 3.759 | 6.61E-12 | 1.20E-08 |
| GO:0022604 | regulation of cell morphogenesis | 473 | 6.8123 | 4.404 | 9.31E-12 | 1.26E-08 |
| GO:0031344 | regulation of cell projection organization | 674 | 9.7072 | 3.709 | 9.72E-12 | 1.26E-08 |
| GO:0031175 | neuron projection development | 940 | 13.538 | 3.176 | 1.23E-11 | 1.39E-08 |
| GO:0030182 | neuron differentiation | 1313 | 18.91 | 2.697 | 4.67E-11 | 4.72E-08 |
| GO:0000902 | cell morphogenesis | 982 | 14.143 | 2.97 | 1.81E-10 | 1.55E-07 |

32˚C (S1 Table). Enrichment analysis of GO terms showed that differentially expressed genes were associated, as expected, with biological processes that are common during embryonic development. Interestingly, genes over-expressed at 26˚C were more frequently associated with neuron development (Table 1), whereas genes over-expressed at 32˚C were mostly associated with muscle development (Table 2).

Although we did not detect thermal-induced sex reversal in *B. vittatus* within the 26˚C–32˚C temperature range, we examined the differentially expressed transcripts in further detail. We found that specific isoforms of *JARID2* and *KDM6B* were over-expressed at 26˚C (Table 3; S1 Table). *JARID2* was the gene with the highest expression difference between 26˚C and 32˚C (Log-FoldChage = -5.2891, FDR = 8.96E-12; Table 3). For both genes, the isoforms that were differentially expressed appeared to have retained an intron. In *JARID2*, the retained intron corresponds to intron 15, the third to last intron (Fig 3), which is the same retained intron that was previously reported for a turtle *T. scripta*, a crocodile *A. mississippiensis*, and the Central Bearded Dragon [16]. In contrast, *KDM6B* retained the last intron, intron 18, instead of the second to last that was reported in the other species [16]. Careful examination of *KDM6B* using BLASTn alignments indicated, contrary to *JARID2*, that the isoform did not retain the full sequence of intron 18. Instead, three shorter sections of intron 18 were included in the *KDM6B* isoform (Fig 3).

## All embryos had hemipenes; Sex has to be established through histology of gonads or genotypic analysis

It should be noted that 100% of the embryos presented intrusive organs (hemipenes), regardless of their genotype (presence/absence of the Y chromosome) or whether the embryos

**Table 2. GO terms enrichment of genes over-expressed at 32˚C.**

| Gene Set | Description | Size | Expect | Ratio | P Value | FDR |
|---|---|---|---|---|---|---|
| GO:0003012 | muscle system process | 423 | 3.1984 | 8.754 | 0 | 0 |
| GO:0006936 | muscle contraction | 339 | 2.5633 | 9.753 | 0 | 0 |
| GO:0030239 | myofibril assembly | 67 | 0.5066 | 25.66 | 2.44E-15 | 7.40E-12 |
| GO:0010927 | cellular component assembly involved in morphogenesis | 106 | 0.80149 | 17.47 | 5.62E-14 | 1.28E-10 |
| GO:0031032 | actomyosin structure organization | 184 | 1.3913 | 11.5 | 6.27E-13 | 1.14E-09 |
| GO:0097435 | supramolecular fiber organization | 640 | 4.8392 | 5.373 | 1.61E-12 | 2.44E-09 |
| GO:0055001 | muscle cell development | 166 | 1.2552 | 11.95 | 1.97E-12 | 2.56E-09 |
| GO:0030240 | skeletal muscle thin filament assembly | 14 | 0.10586 | 66.13 | 3.92E-12 | 4.46E-09 |
| GO:0055002 | striated muscle cell development | 153 | 1.1569 | 12.1 | 9.45E-12 | 9.54E-09 |
| GO:0014866 | skeletal myofibril assembly | 16 | 0.12098 | 57.86 | 1.29E-11 | 1.17E-08 |

**Table 3. Estimated read counts for *JARID2* and *KDM6B* (differentially expressed isoforms).**

|  | FoldChange | FDR | embryos at 32˚C | | embryos at 26˚C | | | |
|---|---|---|---|---|---|---|---|---|
|  |  |  | rep1 | rep2 | rep1 | rep2 | rep3 | rep4 |
| *JARID2* | -5.2891 | 8.96E-12 | 307 | 380 | 3635 | 2143 | 8742 | 5135 |
| *KDM6B* | -3.3684 | 0.000159 | 150 | 125 | 1305 | 933 | 5716 | 1964 |

carried testes or ovaries. The presence of copulatory organs was observed from stage 35 (Fig 4A and 4D). The morphology in this stage corresponded to a prominent lobe that becomes slightly larger. The hemipenes became bilobed in stage 36, with blood supply throughout the entire phallus (Fig 4B and 4E). By stage 37, hemipenes were bifurcated and bilobed with blood supply only in the apical part (Fig 4C and 4F).

## Discussion

Our work examined for the first time to our knowledge the development of gonad during stages 35–37 in a member of the Casque-Headed Lizards family (*Corytophanidae*). We found that gonads were well-differentiated and increased in size and cellular density during these stages. Intriguingly, we found that all embryos showed hemipenes during stages 35–37. This result suggests heterochrony in the regression of hemipenes in females (i. e., both gonad and genital structures development may not be closely coordinated), as it occurs in just three phylogenetically distant lizards [32–34].

Casque-Headed Lizards is the only group of pleurodonts that transitioned from the ancestral XY system to a more recent pair of XY chromosomes. The evolutionary forces that triggered this transition have remained unclear. Here, we tested whether *B. vittatus* showed temperature-dependent sex reversal and whether we could detect genetic signatures of an ancestral TSD system; having a latent sex determination system could facilitate transitions between sex determination systems. We contrasted genotypic data and histological data of embryos from stages 35–37 incubated at three different temperatures. We found no effect of incubation temperatures on the development of gonads: embryos with a Y chromosome

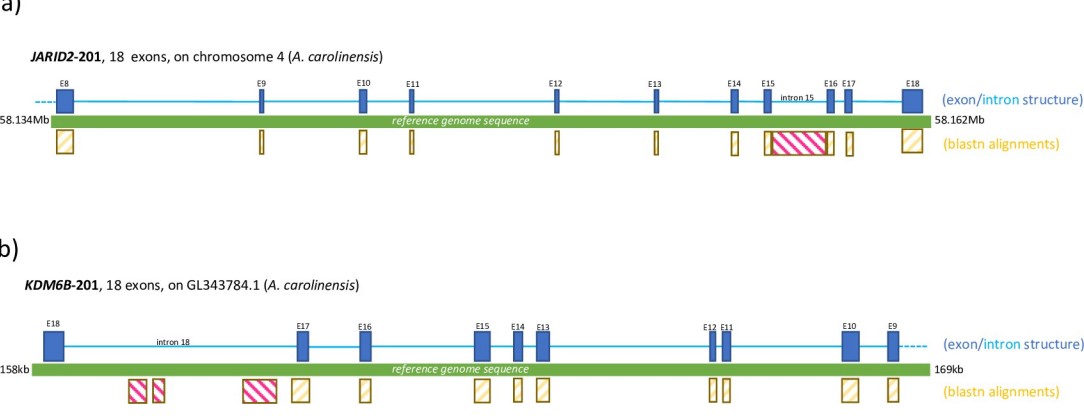

**Fig 3. Exon and intron structure of differentially expressed isoforms of *JARID2* and *KDM6B*.** a) Diagram of the exon/intron structure of the *JARID2* isoform from *B. vittatus* that is over-expressed at 26˚C. Exons are shown as dark blue rectangles, introns are shown as light blue lines. BLASTn matches to exonic sequences are shown as yellow rectangles, whereas matches to intronic sequences are indicated by pink rectangles. The green bar represents the reference genome of *A. carolinensis*. BLASTn alignments were performed against this reference genome. b) Same as in a) but for the *KDM6B* isoform from *B. vittatus* that is over-expressed at 26˚C.

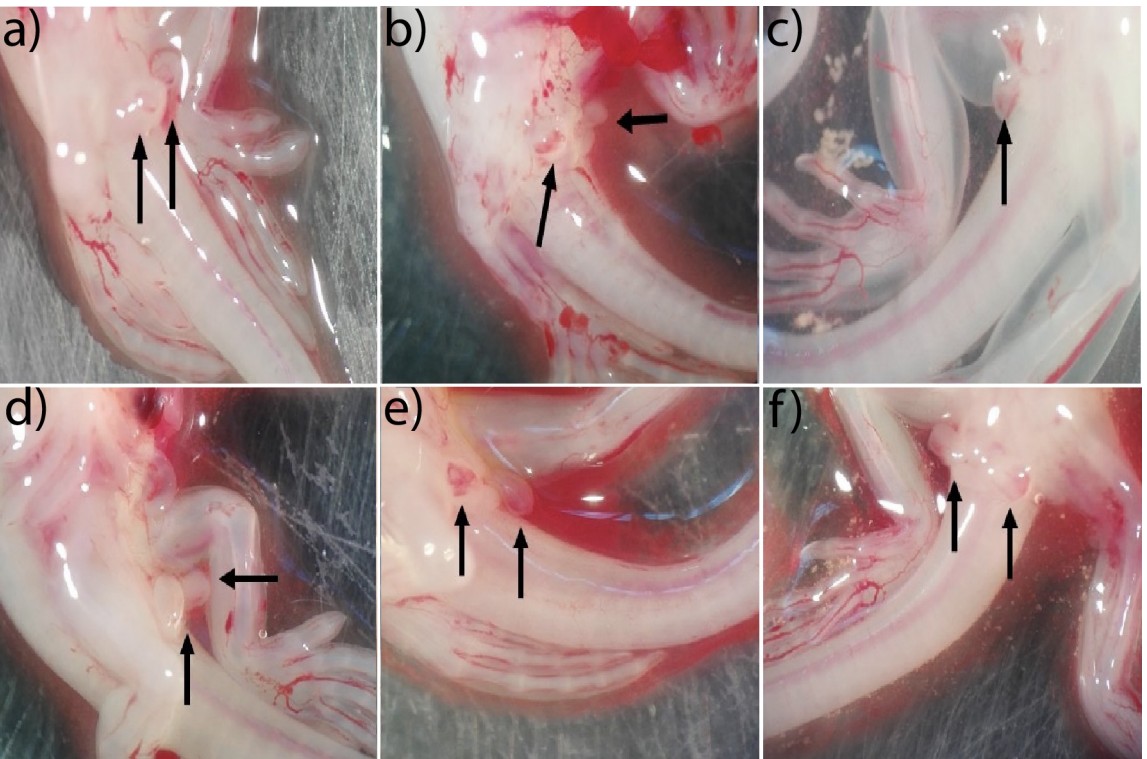

**Fig 4. Hemipenes are present in both male and female embryos.** Females showing hemipenes in a) stage 35, b) stage 36, c) and stage 37. Males showing hemipenes in d) stage 35, e) stage 36, and f) stage 37. Black arrows point at the hemipenes.

developed testes, whereas embryos without a Y chromosome developed ovaries. Thus, *B. vittatus* does not appear to show temperature-dependent sex reversal within the 26–32°C temperature range. We selected these incubation temperatures because they are common male-producing or female-producing temperatures in non-avian reptiles and because they represent average temperatures during the reproductive season of *B. vittatus* [35].

Remarkably, we observed that *JARID2* was the gene with the highest expression difference between low and high incubation temperatures. We found that at a lower temperature (26°C) *JARID2* retained the same intron that was previously reported for a turtle, a crocodile, and the Central Bearded Dragon at male-producing temperatures [16]. Our results suggested that the alternative splicing of *JARID2* is a common trait across non-avian reptiles and appears to be a reminiscence of an ancestral response to temperature [36]. We showed that the new pair of XY chromosomes are dominant; however, it is not clear whether the two XY systems co-existed or whether a potentially latent TSD system facilitated the transition. Future work could explore whether more extreme incubation temperatures (*i. e.*, 36°C, as in the agamid the Central Bearded Dragon) may trigger sex reversal and the nature of the signaling cascades controlled by the temperature-specific isoforms of *JARID2*.

In contrast, the differentially expressed isoform of *KDM6B* did not show the expected pattern since it retained shorter sections of the last intron rather than the complete sequence of the second to last intron. This pattern is rather consistent with *KDM6B* showing alternative 3'-UTR and could indicate that only the *JARID2*-dependent pathway is still sensitive to temperature changes in pleurodonts. This represents additive knowledge to the GSD-TSD continuum and the differentially expressed genes along this continuum may shed light on the evolution of specific signaling pathways during GSD to TSD changes and vice versa.

Moreover, differential expression analyses showed that genes related to neuron development were over-expressed at lower temperatures, whereas genes related to muscular development were over-expressed at higher incubation temperatures. Embryonic development is boosted at elevated incubation temperatures, however, we found that tissues responded differently to temperature. This could ultimately play a role in future sex-related differences in hatchlings and adults, including secondary sexual characters and characteristics. However, the exact role of temperature on non-sex determining characteristics needs further investigation. Although this study provides insights into the patterns of evolution and transitions between GSD and TSD, future studies on this topic are necessary. Our results suggest some variation in gene specific expression of *JARID2*, which has been previously noted to play a role in the TSD pathway [16]. Further work on additional species could help determine whether the *JARID2*-dependent pathway represents the ancestral state in reptiles. Additionally, even though we did not find temperature influence in sex determination at the range of 26–32˚C, we cannot exclude at the moment whether lower or higher incubation temperatures could trigger sex reversal in Casque-Headed Lizards. Additional experiments will be required to explore this possibility.

## Supporting information

**S1 Table. Differentially expressed genes found in this study.**
(XLSX)

## Acknowledgments

We thank Programa Investigadoras e Investigadores COMECYT Edoméx por la cátedra a G. S.-V. We give special thanks to Ana E. López-Moreno, Ailed Pérez-Pérez and Orlando Suárez-Rodríguez for aiding in the collection of lizards. We also thank the community of La Selva del Marinero, state of Veracruz, México, for field assistance. To Justin L. Rheubert for useful comments on a draft of our manuscript.

## Author Contributions

**Conceptualization:** Gabriel Suárez-Varón, Diego Cortez, Oswaldo Hernández-Gallegos.

**Formal analysis:** Gabriel Suárez-Varón, Diego Cortez, Oswaldo Hernández-Gallegos.

**Funding acquisition:** Diego Cortez, Oswaldo Hernández-Gallegos.

**Investigation:** Gabriel Suárez-Varón, Eva Mendoza-Cruz, Armando Acosta, Maricela Villagrán-Santa Cruz, Diego Cortez, Oswaldo Hernández-Gallegos.

**Methodology:** Gabriel Suárez-Varón, Eva Mendoza-Cruz, Armando Acosta, Maricela Villagrán-Santa Cruz, Diego Cortez, Oswaldo Hernández-Gallegos.

**Project administration:** Diego Cortez, Oswaldo Hernández-Gallegos.

**Resources:** Gabriel Suárez-Varón, Diego Cortez, Oswaldo Hernández-Gallegos.

**Software:** Diego Cortez.

**Supervision:** Maricela Villagrán-Santa Cruz, Diego Cortez, Oswaldo Hernández-Gallegos.

**Validation:** Eva Mendoza-Cruz, Armando Acosta, Maricela Villagrán-Santa Cruz, Diego Cortez, Oswaldo Hernández-Gallegos.

**Visualization:** Gabriel Suárez-Varón, Maricela Villagrán-Santa Cruz, Diego Cortez, Oswaldo Hernández-Gallegos.

**Writing – original draft:** Gabriel Suárez-Varón, Diego Cortez, Oswaldo Hernández-Gallegos.

**Writing – review & editing:** Eva Mendoza-Cruz, Armando Acosta, Maricela Villagrán-Santa Cruz.

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
