## [Decision Letter · Decision Letter 0]

28 Feb 2022

PONE-D-22-02482A study of thermally-induced sex reversal in casque-headed lizardsPLOS ONE

Dear Dr. Cortez,

Thank you for submitting your manuscript to PLOS ONE. After careful consideration, we feel that it has merit but does not fully meet PLOS ONE’s publication criteria as it currently stands. Therefore, we invite you to submit a revised version of the manuscript that addresses the points raised during the review process. Both reviewers were enthusiastic about the topic and the potential importance of the findings.  However both authors found issues with the text.  James Walker volunteered to be identified, and made a number of editorial suggestions.  You can find a document in the PLOS One editorial system with his suggest edits (the final version that he sent to me is labeled JWalker edits).  These are only meant to be taken as suggestions, and I leave it to your discretion which of them you wish to implement.  However, many of his suggestions seem sound to me so please do carefully consider his feedback.

The other reviewer identified more substantial issues, particularly areas where the clarity of the manuscript could be improved.  Please do respond directly to each of the issues raised by this reviewer in your rebuttal letter.

We look forward to receiving your revised manuscript.

Kind regards,

Patrick R Stephens, Ph.D.

Academic Editor

PLOS ONE

Journal Requirements:

Reviewers' comments:

Reviewer's Responses to Questions

**Comments to the Author**

1. Is the manuscript technically sound, and do the data support the conclusions?

Reviewer #1: Partly

2. Has the statistical analysis been performed appropriately and rigorously? 

Reviewer #1: Yes

3. Have the authors made all data underlying the findings in their manuscript fully available?

Reviewer #1: Yes

4. Is the manuscript presented in an intelligible fashion and written in standard English?

Reviewer #1: Yes

5. Review Comments to the Author

Reviewer #1: The authors present an interesting paper bringing new insight into the sex determination system of the brown basilisk. The paper is clearly written and well presented. In an ideal world, the sample sizes would be larger, but I appreciate the difficulties associated with using wild caught animals.

Due to the small sample sizes and limited temperature range the available eggs could be incubated at, this study was unable to definitively conclude whether or not the species has sex reversal. As such, I strongly suggest that the title be changed to more accurately reflect the findings of the study. It currently reads like sex reversal was shown to occur in this species, and the short title "Lack of sex reversal in casque-headed lizards" is actually what was found.

I think that it is likely that this species does has sex reversal, and I would be keen to see this expanded on in the discussion. Currently only two other reptiles have been definitively shown to posses sex reversal, and only one of these has an XY system. Both are Australian lizards, so from an evolutionary perspective, sex reversal occurring in a New World species is very interesting.

I noticed that 82 eggs were incubated, but only 48 across the three incubation temperatures reached the target developmental stages. That seems like quite a high mortality rate - are the authors able to comment on this? Was the mortality observed at a particular temperature, were perhaps some of the eggs not actually viable when they were initially incubated? The temperatures aren't particularly extreme, so I would be surprised to see temperature specific mortality. If sex specific mortality has occured this could affect the results presented in the manuscript.

My major concern with the manscript is the tissue used for the RNA-seq analysis. As the methods are currently written I am not entirely clear what tissue was used. Under the "laboratory conditions" section of the methods the posterior part of the embryo was used for histology, so was clearly the part of the embryo containing the gonads. Then the rest of the embryo was preserved for further genetic analysis. This is quite a large part of the embryo containing a wide variety of different tissue types. Then in the "RNA extraction and RNA-seq analysis" section samples were obtained from whole embryos. Were they sub sampled from the other half of the embryo, or were these whole embryos that weren't also used for histology? Regardless, what was the actual tissue type that was sequenced? Or was a large part of the embryo homogenised?

This information is important not only for the methodological clarity of this manuscript, but also because the RNA-seq data has been deposited publicly on SRA, so any other researchers looking to use this data need to have a clear understanding of what the actual sample was. An additional concern is if the same tissue type wasnt used, this significantly affects the validity of the differential gene expression analysis. Based on the low number of differentially expressed genes, I assume that the tissues were consistent between samples, but this does need to be explictly stated.

The discussion feels a little unfinished, and would benefit from a short concluding paragraph, particularly highlighting future research directions, or how it might be possible to definivitely show whether or not the species has sex reversal.

Overall, it is a well presented manuscript, and would be suitable for publication in PLoS One if the concerns outlined above are addressed.

6. PLOS authors have the option to publish the peer review history of their article (what does this mean?). If published, this will include your full peer review and any attached files.

Reviewer #1: No

---

## [Author Response · Author response to Decision Letter 0]

28 Mar 2022

10 March 2022

Patrick R Stephens, Ph.D.

Academic Editor

PLOS ONE

Dear Dr. Stephens,

The authors would like to thank you and the reviewers for their thoughtful insights into our manuscript. These comments and suggestions have largely improved the quality of our work. We have addressed all of the comments in the manuscript (of Reviewer#1 and edits of J. Walker). A detailed list of the corrections is shown below (comments of Reviewer#1). The newly revised version of the manuscript includes the editor and reviewers’ comments. Throughout the entire manuscript, several paragraphs were rewritten for better understanding, and the discussion was expanded.

Academic Editor’s comments:

Both reviewers were enthusiastic about the topic and the potential importance of the findings. However, both authors found issues with the text. James Walker volunteered to be identified and made a number of editorial suggestions. You can find a document in the PLOS One editorial system with his suggest edits (the final version that he sent to me is labeled JWalker edits). These are only meant to be taken as suggestions, and I leave it to your discretion which of them you wish to implement. However, many of his suggestions seem sound to me so please do carefully consider his feedback.

The other reviewer identified more substantial issues, particularly areas where the clarity of the manuscript could be improved. Please do respond directly to each of the issues raised by this reviewer in your rebuttal letter.

Answer:

We thank the Academic Editor for these comments. We have now carefully studied the suggestions made by Dr. Walker. We have accepted most of these suggestions as they clearly improved the text. We have also clarified the questions raised by Dr. Walker regarding the use of some words, the meaning of some sentences, and the common names or families of reptiles we described. Additionally, we have addressed the comments raised by Reviewer#1 regarding the lack of important information in the Methods and the Discussion. Please find below our point-by-point answer to Reviewer#1 comments. We addressed Dr. Walker suggestion’s in the marked-up copy of your manuscript.

Reviewer#1’s comments:

1) Reviewer#1:

Due to the small sample sizes and limited temperature range the available eggs could be incubated at, this study was unable to definitively conclude whether or not the species has sex reversal. As such, I strongly suggest that the title be changed to more accurately reflect the findings of the study. It currently reads like sex reversal was shown to occur in this species, and the short title "Lack of sex reversal in casque-headed lizards" is actually what was found.

Answer:

We thank Reviewer#1 for the detailed revision of our work. Following her/his advice, we have changed the title of the study to reflect the main findings. The new title is: “Genetic determination and JARID2 over-expression in a thermal incubation experiment in Casque Headed Lizard”. Page 1 in the revised version of the manuscript.

2) Reviewer#1:

I noticed that 82 eggs were incubated, but only 48 across the three incubation temperatures reached the target developmental stages. That seems like quite a high mortality rate - are the authors able to comment on this? Was the mortality observed at a particular temperature, were perhaps some of the eggs not actually viable when they were initially incubated? The temperatures aren't particularly extreme, so I would be surprised to see temperature specific mortality. If sex specific mortality has occurred this could affect the results presented in the manuscript.

Answer:

We thank Reviewer#1 for this comment. We agree that the number of eggs that were incubated and the number of eggs reported for the histological/genotypic analyses do not match. We have now corrected this mistake. We would like to clarify that of the 22 females, a total of 130 eggs were obtained. These eggs were randomly assigned to three incubation temperatures (26, 29 and 32°C). Of these, 42 eggs (32%) became contaminated with fungal infection and the embryos died with no effect of temperature on mortality rates (X2 > 0.05). Of the 88 eggs that successfully reached the target developmental stages, 40 were used in a parallel study that aimed to establish the effect of incubation temperatures (26, 29 and 32°C) on the development of the embryos: Suárez-Varón. etal. (2021). REVISTA MEXICANA DE BIODIVERSIDAD. 92:923795 (http://rev.mex.biodivers.unam.mx/index.php/es/variacion-del-estadio-embrionario/). Thus, we used 48 eggs to study the association between incubation temperature, genotype, and the histology of gonads (present study). We have added more information to the Methods. Page 5, first paragraph.

3) Reviewer#1:

My major concern with the manuscript is the tissue used for the RNA-seq analysis. As the methods are currently written I am not entirely clear what tissue was used. Under the "laboratory conditions" section of the methods the posterior part of the embryo was used for histology, so was clearly the part of the embryo containing the gonads. Then the rest of the embryo was preserved for further genetic analysis. This is quite a large part of the embryo containing a wide variety of different tissue types. Then in the "RNA extraction and RNA-seq analysis" section samples were obtained from whole embryos. Were they sub sampled from the other half of the embryo, or were these whole embryos that weren't also used for histology? Regardless, what was the actual tissue type that was sequenced? Or was a large part of the embryo homogenised?

Answer:

We thank Reviewer#1 for this comment. We agree that the Methods are not clear about the tissues used for the analyses. Embryos that reached the target developmental stages were divided into two segments. An upper segment that comprised the head and eyes. And a lower segment where the gonads were located. Each individual was assigned a number so we could match the genetic material versus the histology of the gonads. The upper (head/eyes) segment was then divided longitudinally into two fragments of equal size. One that was homogenized and from which we extracted DNA for the genotypic verification of the Y chromosome, and a second fragment that was also homogenized and from which we attempted to purify RNA; RNA was degraded in many samples though. We have added more information to the Methods. Page 5, first and last paragraphs; Page 6, second paragraph.

4) Reviewer#1 comment:

This information is important not only for the methodological clarity of this manuscript, but also because the RNA-seq data has been deposited publicly on SRA, so any other researchers looking to use this data need to have a clear understanding of what the actual sample was. An additional concern is if the same tissue type wasnt used, this significantly affects the validity of the differential gene expression analysis. Based on the low number of differentially expressed genes, I assume that the tissues were consistent between samples, but this does need to be explictly stated.

Answer:

Reviewer#1 is correct. Using different tissues to conduct differential gene expression analysis may have led to potentially odd results. Besides, detailed information about the samples used is important to replicate our results. As explained above, we used half of the upper segment (head/eyew) of the embryos for DNA/RNA extractions. Tissues were homogenized prior to the purification of the genetic material. We have added more information to the Methods. Page 5, first and last paragraphs; Page 6, second paragraph.

5) Reviewer#1 comment:

The discussion feels a little unfinished, and would benefit from a short concluding paragraph, particularly highlighting future research directions, or how it might be possible to definivitely show whether or not the species has sex reversal.

Answer:

We thank Reviewer#1 for this comment. We have added a new paragraph at the end of the Discussion where we describe the limitations of the work and future research directions. Page 12, second paragraph.

Please do not hesitate to contact us if you need any further information or clarification.

Diego Cortez, PhD 

Centro de Ciencias Genómicas 

UNAM 

Ave. Universidad s/n. CP-62210. Cuernavaca. México

email: dcortez@ccg.unam.mx

Oswaldo Hernández, PhD 

Laboratorio de Herpetología

Universidad Autónoma del Estado de México

Instituto Literario #100 Centro. CP-50000. Toluca. México

email: ohg@uaemex.mx

---

## [Decision Letter · Decision Letter 1]

20 May 2022

PONE-D-22-02482R1Genetic determination and JARID2 over-expression in a thermal incubation experiment in Casque-Headed LizardPLOS ONE

Dear Dr. Cortez,

Thank you for submitting your manuscript to PLOS ONE. After careful consideration, we feel that it has merit but does not fully meet PLOS ONE’s publication criteria as it currently stands. Therefore, we invite you to submit a revised version of the manuscript that addresses the points raised during the review process. The manuscript needs a few additional minor revisions for polish and readability, but overall this is a really solid piece of work. This should be ready to go after a final small revision.  Great work.    

We look forward to receiving your revised manuscript.

Kind regards,

Patrick R Stephens, Ph.D.

Academic Editor

PLOS ONE

Journal Requirements:

Additional Editor Comments (if provided):

My apologies for how long this decision took. Since we were unable to obtain a second review in a timely fashion I have acted as the second reviewer for this revision. I think there are a few minor revisions you still need to make, but overall this is a really solid work. Once you make a few small changes to enhance readability, I believe this will be ready to go. Here are my additional suggestions:

Line 58:

Change “their sex determination mechanisms remain poorly understood”

To “the factors that drive variation in sex determination mechanisms among species remain poorly understood”

Your next sentence demonstrates that we know a lot about how sex determination works in squamates in general, making the first sentence seem like an exaggeration as written.

Line 76:

Delete “however”

Should read:

. . .chromosomes, non-avian reptiles have . . .

Methods:

PLOS is pretty flexible about formatting. However, for readability the formatting should be internally consistent. I suggest that you follow the formatting you use in the Results for headings and subheading throughout.

Line 112: bold the heading like in the discussion

Line 121: there should be a line of space in between 121 and 122, bold the heading

Line 141: bold the heading

Line 148: bold the heading

Line 165: bold the heading

Line 261:

By “top-second gene” do you mean the gene with highest expression difference between 26°C and 32°C? If that is the case, I would delete the phrase “top-second” as I find it really unclear.

I’m also not sure what’s “remarkable” about this result from the intro. In this context “remarkably” implies a result running counter to what most people would expect or that is unusual or unexpected in some other way. However, from what we have read up to this point it’s not clear why you think the result is remarkable.

I would change this sentence to

“JARID2 was the gene with the highest expression difference between 26°C and 32°C . . .”

In the discussion, where you immediately explain what’s remarkable about the result, I think it’s fine to use this phrasing.

Line 322: I would delete “top-second” here as well. I find the phrase really unclear.

Change the sentence to:

“Remarkably, we observed that JARID2 was the gene with the highest expression difference between low and high incubation temperatures.”

Reviewers' comments:

Reviewer's Responses to Questions

**Comments to the Author**

1. If the authors have adequately addressed your comments raised in a previous round of review and you feel that this manuscript is now acceptable for publication, you may indicate that here to bypass the “Comments to the Author” section, enter your conflict of interest statement in the “Confidential to Editor” section, and submit your "Accept" recommendation.

Reviewer #1: All comments have been addressed

2. Is the manuscript technically sound, and do the data support the conclusions?

Reviewer #1: Yes

3. Has the statistical analysis been performed appropriately and rigorously? 

Reviewer #1: Yes

4. Have the authors made all data underlying the findings in their manuscript fully available?

Reviewer #1: Yes

5. Is the manuscript presented in an intelligible fashion and written in standard English?

Reviewer #1: Yes

6. Review Comments to the Author

Reviewer #1: The ammendments the authors have made to the manuscript have addressed my questions, and the manuscript is acceptable for publication in PLoS One

7. PLOS authors have the option to publish the peer review history of their article (what does this mean?). If published, this will include your full peer review and any attached files.

Reviewer #1: No

---

## [Author Response · Author response to Decision Letter 1]

14 Jun 2022

My apologies for how long this decision took. Since we were unable to obtain a second review in a timely fashion I have acted as the second reviewer for this revision. I think there are a few minor revisions you still need to make, but overall this is a really solid work. Once you make a few small changes to enhance readability, I believe this will be ready to go. Here are my additional suggestions:

Line 58:

Change “their sex determination mechanisms remain poorly understood”

To “the factors that drive variation in sex determination mechanisms among species remain poorly understood”

Your next sentence demonstrates that we know a lot about how sex determination works in squamates in general, making the first sentence seem like an exaggeration as written.

A. Done, manuscript was modified to attend this suggestion (line 58).

Line 76:

Delete “however”

Should read:

. . .chromosomes, non-avian reptiles have . . .

A. Done, manuscript was modified to attend this suggestion (line 77).

Methods:

PLOS is pretty flexible about formatting. However, for readability the formatting should be internally consistent. I suggest that you follow the formatting you use in the Results for headings and subheading throughout.

Line 112: bold the heading like in the discussion

A. Done, manuscript was modified to attend this suggestion (line 113).

Line 121: there should be a line of space in between 121 and 122, bold the heading

A. Done, manuscript was modified to attend this suggestion (line 124).

Line 141: bold the heading

A. Done, manuscript was modified to attend this suggestion (line 143).

Line 148: bold the heading

A. Done, manuscript was modified to attend this suggestion (line 150).

Line 165: bold the heading

A. Done, manuscript was modified to attend this suggestion (line 167).

Line 261:

By “top-second gene” do you mean the gene with highest expression difference between 26°C and 32°C? If that is the case, I would delete the phrase “top-second” as I find it really unclear.

A. Done, we delete the phrase “top-second” (line 263).

I’m also not sure what’s “remarkable” about this result from the intro. In this context “remarkably” implies a result running counter to what most people would expect or that is unusual or unexpected in some other way. However, from what we have read up to this point it’s not clear why you think the result is remarkable.

I would change this sentence to

“JARID2 was the gene with the highest expression difference between 26°C and 32°C . . .”

A. Done, manuscript was modified to attend this suggestion (line 262).

In the discussion, where you immediately explain what’s remarkable about the result, I think it’s fine to use this phrasing.

A. Thanks.

Line 322: I would delete “top-second” here as well. I find the phrase really unclear.

Change the sentence to:

“Remarkably, we observed that JARID2 was the gene with the highest expression difference between low and high incubation temperatures.”

A. Done, manuscript was modified to attend this suggestion (line 324).

---

## [Editor Report · Decision Letter 2]

17 Jun 2022

Genetic determination and JARID2 over-expression in a thermal incubation experiment in Casque-Headed Lizard

PONE-D-22-02482R2

Dear Dr. Cortez,

We’re pleased to inform you that your manuscript has been judged scientifically suitable for publication and will be formally accepted for publication once it meets all outstanding technical requirements.

Kind regards,

Patrick R Stephens, Ph.D.

Academic Editor

PLOS ONE

---

## [Editor Report · Acceptance letter]

28 Jun 2022

PONE-D-22-02482R2 

Genetic determination and *JARID2* over-expression in a thermal incubation experiment in Casque-Headed Lizard 

Dear Dr. Cortez:

I'm pleased to inform you that your manuscript has been deemed suitable for publication in PLOS ONE. Congratulations! Your manuscript is now with our production department. 

Kind regards, 

on behalf of

Dr. Patrick R Stephens 

Academic Editor

PLOS ONE